# Teaching Transformers Causal Reasoning Through Axiomatic Training

**Aniket Vashishtha** [1]   **Abhinav Kumar** [2]   **Atharva Pandey** [3]   **Abbavaram Gowtham Reddy** [4]   **Kabir Ahuja** [5]
**Vineeth N Balasubramanian** [6]   **Amit Sharma** [3]

## Abstract

For text-based AI systems to interact in the real world, causal reasoning is an essential skill. Since active interventions are costly, we study to what extent a system can learn causal reasoning from symbolic demonstrations of causal axioms. Specifically, we present an axiomatic training method where the system learns from multiple demonstrations of a causal axiom (or rule), rather than incorporating the axiom as an inductive bias or inferring it from data values. A key question is whether the system would learn to generalize from the axiom demonstrations to more complex scenarios. Our results, based on applying axiomatic training to learn the transitivity axiom and d-separation rule, indicate that such generalization is possible. To avoid data contamination issues, we start with a 67 million parameter transformer model and train it from scratch. On both tasks, we find that a model trained on linear causal chains (along with some noisy variations) can generalize well to complex graphs, including longer causal chains, causal chains with reversed order, and graphs with branching. To handle diverse text inputs, the same method is extended to finetune language models. Finetuning Llama-3.1 8B model on our axiomatic data leads to significant gains on causal benchmarks such as Corr2Cause and CLEAR, in some cases providing state-of-the-art performance surpassing GPT-4.

## 1. Introduction

Causal reasoning can be defined as a set of reasoning procedures consistent with pre-defined axioms or rules that are specific to causality (Galles & Pearl, 1997). For instance, under stable causal models, the transitivity axiom ("if A causes B and B causes C, then A causes C") helps answer questions of cause and effect between pairs of variables in a system. Similarly, the d-separation rule (Pearl, 2009) connects independence of variables and their causal graph structure, and forms the basis of many graph discovery and effect identification algorithms. Given a causal task and data observations from a system, axioms or rules are typically incorporated as inductive biases in a machine learning (ML) algorithm, through regularization, model architecture, or the choice of variables. Depending on the kind of available data—observational, interventional, or counterfactual—Pearl's ladder of causation (Bareinboim et al., 2022) defines the kinds of causal reasoning that is possible.

As axioms are the building blocks of causality, we study whether it is possible to directly learn the axioms or rules using ML models. That is, rather than learning from a dataset sampled from a data-generating process that obeys causal axioms, what if a model can learn an axiom (and thus causal reasoning) directly from symbolic demonstrations of the axiom? This question gains relevance as language models make it possible to learn over symbolic data expressed in natural language. In fact, recent studies have evaluated causal reasoning capabilities of large language models (LLMs) by encoding causal reasoning problems in natural language (Kıcıman et al., 2023; Jin et al., 2023; 2024a). We study whether directly teaching the axioms can be a viable way to improve causal reasoning of language models.

Specifically, we propose a new way to learn causal reasoning through axiomatic training. We posit that causal axioms or rules can be expressed as the following symbolic tuple, ⟨*premise*, *hypothesis*, *conclusion*⟩ where *hypothesis* refers to a causal claim and *premise* refers to any relevant information to decide whether the claim is true or not (*conclusion*). The conclusion could simply be *Yes* or *No*. For example, consider the task of inferring causal relationships from correlational statements in the Corr2Cause dataset (Jin

---

[1]Work primarily done as a Research Fellow at Microsoft Research India, with additional contributions made at UIUC, USA [2]Massachusetts Institute of Technology, USA [3]Microsoft Research, India [4]CISPA Helmholtz Center for Information Security, Germany. Part of the work done at IIT Hyderabad [5]University of Washington, USA [6]Work primarily done at IIT Hyderabad, with additional contributions made at MSR India. Correspondence to: Aniket Vashishtha <aniketv2@illinois.edu>, Amit Sharma <amshar@microsoft.com>.

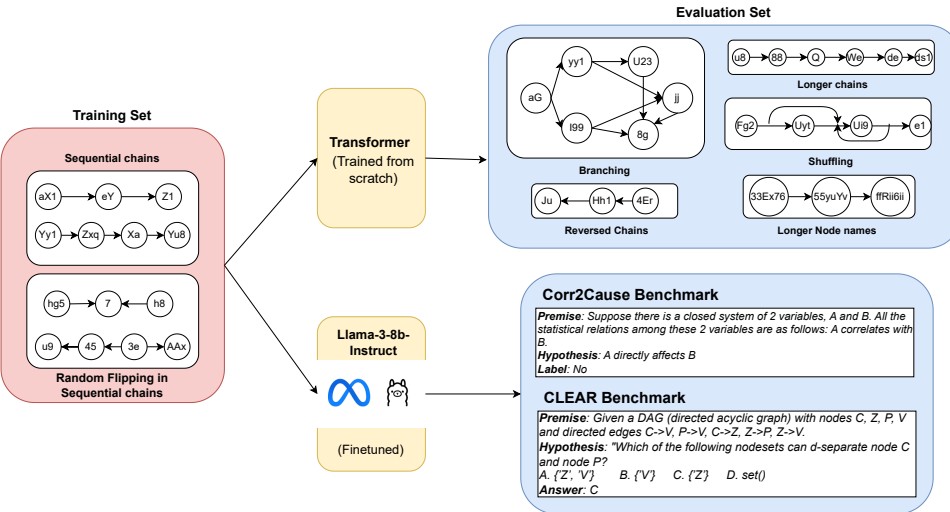

Figure 1: *Axiomatic Training* for imparting causal reasoning to language models. Given an axiom, we construct a training dataset comprising <*premise, hypothesis, conclusion*> triplets based on simple chain-like graphs of 3-6 nodes. A transformer model trained from scratch on such instances generalizes to much more complex graphs, including longer causal chains with >6 nodes, branched networks with higher average in-degree and out-degree, complete reversals, shuffled statements, and longer node names. Moreover, when an existing model such as LLama-3-8B model is trained on the same dataset, it leads to significant (up to 20 p.p.) improvement in accuracy on causal reasoning benchmarks such as CLEAR and Corr2Cause.

et al., 2024a), which we empirically study in this paper. The *premise* can be statements about statistical (in)dependence: *"Premise: T causes T1. T2 causes T. T2 causes t. T2 causes T3. t causes T. t causes T1. t causes T3"*; the hypothesis can be a question about cause-and-effect, *"Are T3 and T d-separated given $(t, T2)$?"*; and the *conclusion* would be *"Yes"*. This tuple is a demonstration of the *d-separation* rule (see Section 3 for definition). Based on this template, our key insight is that a large number of synthetic tuples can be generated, e.g., by changing the variable names, changing the number of variables, changing the order, and so on. The question is: if a model is trained on such data, would it learn to apply the axiom to new, more complex scenarios?

To answer this question, we consider a setup where a model is trained on axiomatic demonstrations over simple chain-like graphs with 3-6 nodes and evaluated on more complex graphs, including longer chain-like graphs with 7-15 nodes, graphs with branching, longer variable names, and edge direction perturbations (see Figure 1). To avoid any contamination concerns with the pre-training data of an existing language model, we first train a transformer model from scratch. For both transitivity and d-separation, we find that a model trained on axiomatic demonstrations learns to apply the axiom multiple times to answer questions over more complex graphs. In particular, diversity in the training data plays a crucial role in enabling this generalization. For transitivity, a model trained on a combined dataset of simple directed chains and chains with some edges randomly

reversed, generalizes well across all kinds of evaluation scenarios. However, a model trained only on simple directed chains fails to generalize to graphs with branching or edge direction perturbations. Moreover, for d-separation, our 67 million parameter model outperforms billion-scale models such as GPT-4 under both zero-shot and multi-shot settings. Extending the findings on positional embedding for length generalization in NLP tasks (Kazemnejad et al., 2023; Bhattamishra et al., 2020; Haviv et al., 2022), we find that rotary positional encoding works the best for causal generalization, closely followed by no positional encoding.

Next, we study whether the same axiomatic training dataset can also help to improve causal reasoning of pre-trained large language models. We fine-tune Llama-3-8B-Instruct model over axiomatic datasets for transitivity and d-separation. We evaluate on two benchmarks of causal reasoning in natural language: CLEAR (Chen et al., 2024) and Corr2Cause (Jin et al., 2024a). The CLEAR dataset tests language models' understanding of causal graphs through a wide range of binary and multi-choice tasks, including d-separation, Markov equivalence, and backdoor adjustment. Even though we don't finetune on their dataset or question type, we find significant performance gains on the d-separation task after axiomatic fine-tuning: accuracy increases from 30 to 70% on *Yes/No task*, and goes from 33 to 50% on *Multi-Choice Questions*.

Corr2Cause is another challenging dataset to assess a lan-

guage model's ability to infer causal relationships (e.g., ancestor, collider, or fork) given correlational statements expressed in natural language. Solving this task requires knowledge of multiple causal concepts including d-separation, transitivity, Markov property and the Markov equivalence class. Jin et al. (2024a) show how frontier models such as GPT-4 also fail at this task. We show that axiomatic finetuning leads to a substantial F1 improvement of nearly 20 percentage points over the base Llama3 model and even surpasses GPT-4, highlighting the effectiveness of axiomatic training for causal reasoning on complex datasets.

Overall, our work provides a new paradigm of teaching models causal reasoning through symbolic demonstrations of axioms or rules, which we call *axiomatic training*. Such symbolic data can be cheaply generated for multiple axioms and added to the finetuning data of language models. More generally, our results using axiomatic data contribute to the literature on causal learning from *passive data* (Lampinen et al., 2023). Our code repository can be accessed at: https://github.com/AniketVashishtha/Causal_Axioms.

## 2. Related Work

**LLMs for Knowledge-Driven Causal Reasoning:** Recent developments in Large Language Models (LLMs) have highlighted their potential for knowledge-driven causal discovery. Unlike traditional methods which focus on statistical patterns or correlations, LLMs utilize knowledge acquired through their pretraining to reason about and identify causal structures based on metadata of variables (Kıcıman et al., 2023; Ban et al., 2023; Long et al., 2023; Willig et al., 2022; Vashishtha et al., 2025). However, possibility of memorization of existing benchmarks in the pretraining of these LLMs has been a major criticism. As a result, recent work (Zečević et al.) argues that LLMs are not actually performing causal reasoning, but simply learning correlations about causal facts. In addition, there are critical failure modes of using LLMs for causal discovery due to hallucinations or not obeying the acyclic constraint when generating graph edges (Vashishtha et al., 2025). To evaluate causal reasoning capabilities of LLMs, (Jin et al., 2024a) and (Jin et al., 2023) propose formal causal inference evaluation benchmarks to infer direct and indirect causal relationships, and highlight the failure of LLMs in performing accurate causal reasoning.

**Impact of Positional Encoding on Generalization:** Length generalization capabilities of transformers has been studied in the past to better understand their different failure modes across various settings (Hupkes et al., 2020; Zhang et al., 2023; Furrer et al., 2021). Previous work (Kazemnejad et al., 2023; Bhattamishra et al., 2020; Haviv et al., 2022; Shen et al., 2023) emphasizes the impact of positional

encoding in length generalization capability of transformers. To understand how transformers can be optimized for learning through axiomatic training and generalizing to unseen larger causal structures, we also examine different types of positional encoding such as no positional encoding (PE), Learnable PEs (Radford et al., 2018) and Sinusoidal PEs (Vaswani et al., 2017).

**Synthetic data generation for teaching transformers reasoning:** Synthetic data generation has been explored for optimising model training for reasoning. For example, (Li et al., 2023; Gunasekar et al., 2023) use LLM-generated synthetic text for training Phi-1 and Phi-1.5 models and show impressive performance for reasoning-based tasks. (Trinh et al., 2024) introduce a novel neuro-symbolic framework to pre-train a transformer model for Olympiad-level math problems. (Morishita et al., 2024) construct synthetic training data to improve language models' performance on logical reasoning tasks. Building on this stream of work, we apply synthetic data generation for teaching causal reasoning.

## 3. Preliminaries: Causal Axioms and Rules

Instead of performing causal reasoning using observational or interventional data, we study whether it is possible to learn general rules of causality directly from symbolic axioms. There has been fundamental work from Galles & Pearl (1997) where they axiomatize causal relevance (or equivalently irrelevance). They show that for a given *stable probabilistic* causal model (defined below), there exists a finite set of axioms that are completely characterized by axioms of path interception in corresponding directed graphs. Additionally, causal inference in practice depends on a few key rules, such as d-separation and do-calculus rules. Learning these rules can have a tangible impact on practical causal tasks such as graph discovery and effect inference. While we call the method axiomatic training, we consider learning both causal axioms and rules. Throughout this work, we assume no unobserved confounders.

**Notation.** We denote a random variable with an uppercase letter (e.g., $X, Y, Z$) and use lowercase letters (e.g., $x, y, z$) to denote the values taken by the corresponding random variable, written as $X = x, Y = y, Z = z$. We represent the probability of a random variable $X_i$ as $\mathbb{P}(X_i)$. Let $\mathcal{G}(\mathbf{X}, \mathbf{E})$ be a directed acyclic graph (DAG) consisting of a set of variables $\mathbf{X} = \{X_1, \ldots, X_n\}$ and a set of directed edges $\mathbf{E}$ among variables in $\mathbf{X}$. Let $pa(X_i) = \{X_k | X_k \rightarrow X_i\}$, $de(X_i) = \{X_k | X_k \leftarrow \cdots \leftarrow X_i\}$, $ch(X_i) = \{X_k | X_i \rightarrow X_k\}$ denote the set of *parents*, *descendants* and *children* of $X_i$ respectively. Given two nodes $X_i, X_j$ we call a third node $X_k$ as a *collider* if both $X_i$ and $X_j$ are parents of $X_k$.

### 3.1. Axioms of Causality: Transitivity

**Definition 3.1** (**Causal Irrelevance**, adapted from Defn. 7 in (Galles & Pearl, 1997)). $X$ is probabilistically causally irrelevant to Y given Z, written $(X \nrightarrow Y|Z)$ iff: $\mathbb{P}(y|z, do(X) = x) = \mathbb{P}(y|z, do(X) = x'), \forall x, x', y, z$ i.e., once we hold Z fixed at z, intervening on X will not change the probability of Y.

Next, we restate the stability assumption for a causal model from (Galles & Pearl, 1997) that gives a richer set of finite axiomatization for probabilistic causal irrelevance.

**Assumption 3.2** (Stability, Definition 9 in (Galles & Pearl, 1997)). Let $\mathcal{M}$ be a causal model. Then an irrelevance $(X \nrightarrow Y|Z)$ in $\mathcal{M}$ is stable if it is shared by all possible probability distribution over $\mathcal{M}$. The causal model $\mathcal{M}$ is stable if all of the irrelevances in $\mathcal{M}$ are stable.

Under the stability assumption (see Assumption 3.2), Galles & Pearl (1997) states six axioms that completely characterize causal irrelevance (Definition 3.1) via axioms of path interception in the directed graphs. An axiom of causal irrelevance is of the form (given in conjunctive normal form):

$$\bigwedge_s \bigvee_t (\boldsymbol{X}_i^{s,t} \nrightarrow \boldsymbol{X}_j^{s,t} | \boldsymbol{X}_k^{s,t}) \implies \bigwedge_l \bigvee_n (\boldsymbol{X}_i^{l,n} \nrightarrow \boldsymbol{X}_j^{l,n} | \boldsymbol{X}_k^{l,n})$$

where $\wedge$ is "logical and", $\vee$ is "logical or" and for a given $(s, t)$ or $(l, n)$ pair, $\boldsymbol{X}_i, \boldsymbol{X}_j, \boldsymbol{X}_k$ are disjoint subsets of observed variables $\boldsymbol{X}$. In the above causal irrelevance statement, if the antecedent is true, the consequent is also true.

**Transitivity Axiom.** We illustrate our axiomatic training procedure through the transitivity axiom. Following the stability assumption above, we consider the class of interventional distributions in which the transitivity causal irrelevance axiom holds (Sadeghi & Soo, 2024). Formally, for a stable probabilistic causal model (§3), given variables $X, Y, Z$ in the system, the transitivity axiom is:

$$(X \nrightarrow Y|Z) \implies (A \nrightarrow Y|Z) \vee (X \nrightarrow A|Z)$$
$$\forall A \notin X \cup Z \cup Y$$

which can be simplified using the contrapositive. We call the LHS as *Premise* and the RHS as *Hypothesis*.

$$\exists A \notin X \cup Y \cup Z \ s.t. \quad \underbrace{(X \rightarrow A|Z) \wedge (A \rightarrow Y|Z)}_{P:\text{premise}}$$
$$\implies \underbrace{(X \rightarrow Y|Z)}_{H:\text{hypothesis}} \quad (1)$$

### 3.2. d-separation rule

The d-separation rule connects causal graph structure with conditional independence in $\mathbb{P}(\boldsymbol{X})$.

**Definition 3.3** (Definition 1.2.3 in Pearl (2009)). Given a DAG $\mathcal{G}(\boldsymbol{X}, \boldsymbol{E})$, two sets of random variables $\boldsymbol{X}_i$ and $\boldsymbol{X}_j$ are said to be d-separated by a third set $\boldsymbol{X}_z$ if all the *paths* between $\boldsymbol{X}_i$ and $\boldsymbol{X}_j$ in $\mathcal{G}$ are blocked by $\boldsymbol{X}_z$. A *path* $p$ between $\boldsymbol{X}_i$ and $\boldsymbol{X}_j$ is said to be blocked by a set of nodes $\boldsymbol{X}_z$ iff 1) $p$ contains a fork (i.e., $\cdot \leftarrow A \rightarrow \cdot$) or a chain (i.e., $\cdot \rightarrow A \rightarrow \cdot$) such that the middle node $A$ is in $\boldsymbol{X}_z$, or 2) $p$ contains a collider ($\cdot \rightarrow A \leftarrow \cdot$) such that the middle node $A$ is not in $\boldsymbol{X}_z$ and no descendant of $A$ is in $\boldsymbol{X}_z$.

Given $\mathbb{P}(\boldsymbol{X})$ is Markov with respect to $\mathcal{G}$, if two sets of random variable $\boldsymbol{X}_i$ and $\boldsymbol{X}_j$ are d-separated by $\boldsymbol{X}_z$, then they are conditionally independent of each other given $\boldsymbol{X}_z$.

## 4. Axiomatic Training for Transformers

Given an axiom, our key idea is to generate thousands of synthetic symbolic expressions that can be used to train a transformer on how to use the axiom. The trained model is then evaluated on whether it can apply these axioms to new causal structures that were not available in the training set. Below we describe how we generate the training data and the model architecture details.

### 4.1. Training Data: Diversity is key

As mentioned above, an axiom consists of a tuple, $\langle premise, hypothesis, conclusion \rangle$. Based on the specific axiom, we can map a hypothesis given the premise to its correct label ('*Yes*' or '*No*'). To create a training dataset, we randomly sample a causal DAG $\mathcal{G}$ and enumerate $N$ random tuples of $\{(P, H, L)\}_N$ where $P$ is the premise, $H$ is the hypothesis and $L$ is the label *(Yes/No)*. The premise describes the edges of the graph and is expressed in natural language, e.g., *"X causes Y. Z causes Y.".* Given a premise $P$ based on the causal graph's edges, if the hypothesis can be derived by applying the specified axiom (once or multiple times), then label $L$ is *Yes*; otherwise, *No*. For example, for the transitivity axiom, suppose the underlying true causal graph of a system is a chain, $X_1 \rightarrow X_2 \rightarrow X_3$. Then, the premise will be $X_1 \rightarrow X_2 \wedge X_2 \rightarrow X_3$. A corresponding hypothesis for the transitivity axiom could be $X_1 \rightarrow X_3$ will have label *Yes* whereas another hypothesis $X_3 \rightarrow X_1$ will have label *No*. The former would create a training data instance with the following text, *"X1 causes X2. X2 causes X3. Does X1 cause X3? Yes.".* Note that the axiom can be inductively applied multiple times to generate more complex training tuples. Another possible hypothesis for the *d-separation* rule could be "Are $X_1$ and $X_2$ d-separated given $\{X_3\}$?" and the label will be *No*.

We train the model on data from simple causal graphs such as sequential chains with 3-6 nodes and evaluate its performance on more complex graphs (refer fig. 1). To enhance generalization, we introduce structured perturbations in the

training data across three axes: node names, causal structure types, and the number of nodes in the causal graph.

1. **Node names**: Each node in the graph is represented by an alphanumeric name comprising 1-3 characters. The length of a name and the specific characters are randomly selected during data generation.
2. **Causal Graph Topology**: We consider two main types of causal graphs in the training set.
   (a) **Sequential**: All causal edges are directed forward, forming a typical chain DAG, e.g., X → Y → Z.
   (b) **Random Flipping**: Given a chain of sequential nodes, we randomly reverse some edges eg. $X \rightarrow Y \leftarrow Z$. This can be expressed simply through natural language like: *"X causes Y. Z causes Y."*. This introduces forks and colliders that help add complexity to model training, thus aiding generalization across a larger space of graphs.
3. **Number of nodes in graph**: To facilitate the generalization of transformers over graphs of different sizes we incorporate chains of varying lengths, ranging from 3 to 6 nodes in our training set.

## 4.2. Tokenization, Training Loss & Architecture

We train a decoder-based 67 million parameter model based on GPT-2's architecture. The model has 12 attention layers, 8 attention heads and 512 embedding dimensions. We train the transformer model from scratch to ensure that the model has not seen such axioms in the pertaining step and thus requires a true correct understanding of axioms to perform well. Later we also tested on a pre-trained model which was fine-tuned on our dataset.

**Tokenization.** Since the training dataset follows a specific structure, we develop a custom tokenizer. Alphanumeric node names are tokenized at a character level, while special terms such as *'causes', 'Does', 'cause', 'Yes'*, and *'No'* are tokenized at the word level. Such an approach avoids out-of-vocabulary (OOV) tokens at test time since the alphanumeric node names in the test set can be different than those in the training set. Following this approach, the vocabulary size of our transformer model is 69.

**Loss function.** Given a dataset, the loss function is defined based on the ground truth label for each tuple, represented as $\mathbb{E}_{(P,H,L)\sim\mathbb{P}_{\text{train}}} -\log\mathbb{P}(L|P,H)$. A preliminary analysis indicated promising results with this loss formulation compared to next token prediction loss.

**Positional Encoding.** In addition to tokenization and loss function, recent work (Kazemnejad et al., 2023) shows that the choice of positional encoding is important for generalizing a transformer to longer or complex inputs. Therefore, we evaluate different positional encodings on their impact on generalization in causal tasks: learnable (LPE) (Radford et al., 2018), sinusoidal (SPE) (Vaswani et al., 2017), rotary (RoPE) position encodings (Su et al., 2024), and no positional encoding (NoPE) (Kazemnejad et al., 2023; Haviv et al., 2022). See Appendix F for details.

**Finetuning.** Apart from training a transformer from scratch, we fine-tune a pre-trained language model (Llama-3-8b-Instruct (gra, 2024)) on our axiomatic training data. Finetuning a pretrained model like Llama, allows us to evaluate causal reasoning in a wider variety of contexts, leveraging its expansive vocabulary and the versatility provided by pretraining. While a model trained from scratch on axiomatic data is limited by its narrow vocabulary and structure, finetuning a pretrained model allows direct evaluation on benchmarks like Corr2Cause, which implicitly require the application of the target axioms, thus enabling us to stress test our framework in more diverse settings. Refer J for more details about the finetuning setup.

## 4.3. Evaluation setup: Assessing Axiomatic Learning

We consider two types of evaluation: 1) on synthetic datasets where we directly test the models on axioms and, 2) on existing benchmarks corresponding to different high-level causal tasks where we expect the axioms to be helpful.

**Synthetic evaluation.** To evaluate if a trained model has learned the correct understanding of an axiom instead of shortcuts or correlation-based features, designing an out-of-distribution (OOD) evaluation set is important. We evaluate our model on multiple types of complex graphs that are unseen during training.

1. **Length**: Evaluating whether our model accurately infers causal relationships for chain graphs (both sequential and ones with random flipping) longer than those in the training set. Specifically, we train the model on chains with size 3-6 and evaluate on chains of size 7-15.
2. **Node Name Shift**: Testing the model's performance on longer node names, from 1-3 characters in the training set to 8-10 characters, motivated by Jin et al. (2024a) who show how change in node names can lead to generalization failure on causal tasks for language models.
3. **Order of Chains**: a) **Completely reversed chains**: This evaluation is inspired by the reversal curse (Berglund et al., 2024) that revealed generalization failure of LLMs in answering questions in reversed sequences despite knowing the answers in the original order. We evaluate our model on completely reversed chains, a structure that was not in the training data. A completely reversed chain will be of the form $X \leftarrow Y \leftarrow Z$, written in natural language as: *"Y causes X. Z causes Y."*, where $X, Y, Z$ are replaced by random alphanumeric names. b) **Shuffling of Sequences:** Here we shuffle the order of causal edges presented in each training row to add complexity and break sequential order.

4. **Branching Factor**: One of the most complex evaluation setups, with DAGs containing multiple branches, colliders, and forks. Let the branching factor be defined as the ratio between a number of edges and a number of nodes. Thus, the branching factor for the training set is $\leq 1$. Then, we create a different evaluation set with multiple densely branched networks constructed using the Erdös-Rényi model, with different branching factors.

### 4.4. Benchmark evaluation

To test whether such simple axiomatic training is helpful in more complex scenarios, we evaluate our models on causal reasoning benchmarks, Corr2Cause (Jin et al., 2024a) and CLEAR (Chen et al., 2024). Unlike past methods as in Jin et al. (2024a), we do not finetune on these datasets directly. Instead, we finetune a language model on synthetic axiomatic data and evaluate whether the model generalizes to these external benchmarks. We discuss the results on these benchmarks in Section 7.

### 4.5. Baselines Using Existing LLMs

Given recent work on how LLMs can be leveraged for causal reasoning (Kıcıman et al., 2023; Vashishtha et al., 2025; Ban et al., 2023), we include language models such as GPT-4 *(gpt-4-32k)* (ope, 2024), Gemini *(gemini-pro)* (gem, 2024) and Phi-3 *(Phi-3-mini-128k-instruct)* (abd, 2024) as baselines. Each model is significantly larger than our model and known to perform well on reasoning tasks, with the smallest model Phi-3 having 3.8B parameters (Li et al., 2023). Refer App. B and C.1 for zero and multi-shot prompts used. On evaluating the baselines for the transitivity task, we find that GPT-4 is significantly better. Hence, for the more complex d-separation task, we evaluate only GPT-4.

## 5. Axiomatic Training For Transitivity Axiom

In all our experiments, we consider an empty conditioning set $Z$ for simplicity.

### 5.1. Training and Evaluation Datasets

**Training Datasets.** The training data consists of sequential chains of lengths from [3,6]. In addition to sequential chains, random flipping of edges is done with 0.5 probability. See Appendix G for details on these hyperparameters. Our training data consists of 175k axiom demonstrations. We use three versions of training data to evaluate the impact of different data perturbations.

1. **Only Causal Chains (OCC)**: This set comprises of graphs with only sequential chains (see causal graph topology in Sec 4.1 for details).
2. **Training Setup 1 (TS1)**: This setup comprises of 73k examples where the underlying graphs has random flipping

| Model/Nodes | 3 | 4 | 5 | 6 | 7 | 8 | 9 |
|---|---|---|---|---|---|---|---|
| **Baselines (Zero Shot)** | | | | | | | |
| GPT-4 | 0.99 | 0.97 | 0.89 | 0.85 | **0.95** | **0.90** | 0.90 |
| Gemini Pro | 0.75 | 0.73 | 0.72 | 0.76 | 0.71 | 0.68 | 0.74 |
| Phi-3 | 0.88 | 0.86 | 0.82 | 0.79 | 0.76 | 0.73 | 0.79 |
| **Baselines (Multi Shot)** | | | | | | | |
| GPT-4 | **1.00** | **0.99** | **0.97** | **0.95** | 0.94 | **0.90** | **0.92** |
| Gemini Pro | 0.95 | 0.85 | 0.83 | 0.79 | 0.79 | 0.73 | 0.75 |
| Phi-3 | 0.88 | 0.83 | 0.82 | 0.80 | 0.83 | 0.76 | 0.78 |
| **Axiomatic Training** | | | | | | | |
| TS1 w NoPE | **1.00** | 0.93 | 0.85 | 0.83 | 0.78 | 0.73 | 0.73 |
| TS1 w LPE | **1.00** | 0.93 | 0.87 | 0.83 | 0.79 | 0.74 | 0.73 |
| TS1 w SPE | 0.99 | 0.92 | 0.85 | 0.81 | 0.76 | 0.74 | 0.61 |
| TS1 w RoPE | 1.0 | 0.93 | 0.87 | 0.85 | 0.81 | 0.78 | 0.76 |
| TS2 w NoPE | 0.99 | 0.93 | 0.86 | 0.82 | 0.79 | 0.74 | 0.73 |
| TS2 w LPE | **1.00** | 0.92 | 0.85 | 0.83 | 0.77 | 0.74 | 0.71 |
| TS2 w SPE | 0.99 | 0.94 | 0.86 | 0.81 | 0.76 | 0.72 | 0.64 |
| TS2 w RoPE | 1.0 | 0.95 | 0.89 | 0.86 | 0.82 | 0.79 | 0.76 |
| OCC w RoPE | 0.78 | 0.71 | 0.64 | 0.65 | 0.63 | 0.61 | 0.61 |

Table 1: **Evaluation on MultiEval$_{\text{SLR}}$ dataset.** Accuracy of axiomatically trained models with another baseline on the most complicated setups. For OCC we only report performance with RoPE encodings, which is the best performing setup for this dataset. See Sec 5.2 for details. **Bold** numbers denote the highest value on a test set, while the underlined ones denote the second best.

and 101k causal graphs where the underlying graphs has sequential chains. Since flipping is done randomly across all consecutive pairs of nodes in the given chain, some complete reversals are also formed. In this training set, around 12k completely reversed chains are present.

3. **Training Setup 2 (TS2)**: This setup comprises more simple sequential chains (132k), while we decrease chains with random flipping (42k) to keep the overall size around 175k to see the effect of adding examples with complicated graphs on model's performance.

**Evaluation Datasets.** In addition to the evaluation sets described earlier in Sec 4.3 (length generalization, node name shift, order of chains, and branching), we add another evaluation set that is a combination of three shifts.

**MultiEval$_{\text{SLR}}$ (Shuffling + Random Flipping + Length Sequence)**: This setup merges three kinds of complexities: shuffling of sentence for representing the sequences, each sequence having random flipping, and some sequences having longer length than sequences in training set (upto 9).

### 5.2. Results: Generalizing to Complex Causal Graphs

We train our model using axiomatic training on different kinds of datasets, TS1, TS2, and OCC, with different positional encoding (NoPE, LPE, and SPE) as described in Sec 5.1. Results on all evaluation settings are in Appendix Tables A3, A4 and A5.

| Model/Nodes | 5 | 8 | 10 | 12 |
|---|---|---|---|---|
| ***Baselines (Zero shot)*** | | | | |
| GPT-4 | 0.95 | 0.90 | 0.88 | 0.86 |
| Gemini Pro | 0.74 | 0.76 | 0.73 | 0.71 |
| Phi-3 | 0.83 | 0.79 | 0.77 | 0.80 |
| ***Baselines (Multi shot)*** | | | | |
| GPT-4 | **0.97** | **0.93** | **0.94** | **0.93** |
| Gemini Pro | 0.76 | 0.79 | 0.77 | 0.79 |
| Phi-3 | 0.78 | 0.82 | 0.79 | 0.79 |
| **Axiomatic Training** | | | | |
| OCC w RoPE | 0.72 | 0.68 | 0.66 | 0.62 |
| TS1 w LPE | 0.82 | 0.72 | 0.69 | 0.68 |
| TS1 w SPE | 0.78 | 0.61 | 0.59 | 0.57 |
| TS1 w NoPE | 0.82 | 0.74 | 0.69 | 0.66 |
| TS1 w RoPE | 0.86 | 0.79 | 0.74 | 0.71 |
| TS2 w LPE | 0.80 | 0.72 | 0.67 | 0.64 |
| TS2 w SPE | 0.79 | 0.59 | 0.54 | 0.52 |
| TS2 w NoPE | 0.82 | 0.73 | 0.68 | 0.64 |
| TS2 w RoPE | 0.88 | 0.79 | 0.72 | 0.68 |

Table 2: **Evaluation with branching factor 1.4.** Accuracy of axiomatically trained models with LM baselines on the causal graphs with branching factor 1.4.

**MultiEval$_{SLR}$ Dataset.** We evaluate our models and other baselines on the challenging **MultiEval$_{SLR}$** dataset since it includes example that are different from training dataset in terms of size and complexity of causal graph. Table 1 summarizes the results for this dataset. While GPT-4 performs best, models trained with RoPE position encodings still achieve strong results, surpassing Gemini Pro and Phi-3 in both zero-shot and multi-shot settings for majority of node lengths.

A similar trend is seen for completely reversed sequences (Table A3). This task presents extreme out-of-distribution data, as the training data contains left-to-right edges, while the test data has only right-to-left edges. TS2 (NoPE) consistently outperforms Gemini-Pro and Phi-3, and remains competitive with GPT-4 (zero shot). In particular, its accuracy (0.94 for chains of length 6) is substantially higher than Gemini Pro and Phi-3 (0.71 and 0.75 respectively).

**Branched Causal Graphs.** Even though our models are trained on simpler graphs like sequential and random-flipping, we want to test our models on structurally harder graphs not considered in **MultiEval$_{SLR}$**. To do so, we introduce general Erdos-Renyi graphs as the causal sequences while the training data contains only linear chains. We vary the branching factor of the graph as defined in Sec 4.3 between 1.4 and 2 for all our experiments on graphs with different numbers of nodes. Table 2 summarizes the results of this experiment. While GPT-4 achieves the highest ac-

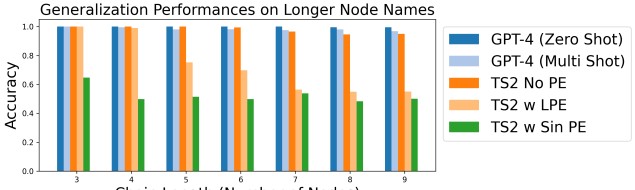

Figure 2: Evaluating generalization on causal sequences (without random flipping) with longer node names (than the ones used in sequences in train set). TS-2 training set with no positional encoding leads to the best performance. Refer table A5 for complete results.

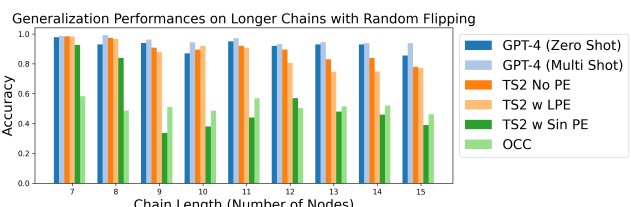

Figure 3: Generalizing to longer unseen causal sequences (>6 nodes) with random flipping on TS2 and OCC (with NoPE) train sets. OCC-trained models struggle due to limited edge-level variability, while TS2 NoPE consistently performs well. Refer table A4 for complete results

curacy as graph sizes increase, our TS1 (RoPE) and TS2 (RoPE) model outperforms Gemini Pro (branching factor 1.4) in for graphs with size 5 and 8 under zero-shot settings. On graphs with 12 nodes and a 1.4 branching factor, TS2 (RoPE) achieves 68% accuracy, far better than random (50%), despite training only on graphs with branching factors ≤ 1. Although LLMs excel in multi-shot settings, our model's performance is comparable even on more complicated causal graphs than the ones they were trained on.

**Further Ablations.** We run further ablations to understand the generalization behavior of our models. In particular, we experiment with generalization to unseen node names (Appendix Table A5), generalization to unseen lengths for graphs with linear chains and random flipping of edges (Appendix Table A4), and generalization to graphs with branching factor of 2 (Tab. A6).

# 6. Axiomatic Training for D-Separation Rule

Similar to the transitivity axiom, we now train our model on instances of the d-separation rule.

## 6.1. Training Dataset and Setup

We follow a similar strategy to generate the training dataset as for the transitivity axiom. The training dataset consists

| Branching (Bfactor = 1.4) | | | | |
|---|---|---|---|---|
| Models | 5 | 8 | 10 | 12 |
| *Baselines* | | | | |
| GPT-4 | 0.53 | 0.544 | 0.62 | 0.52 |
| *Finetuned Results* | | | | |
| Llama-3-8b-Instruct | 0.474 | 0.490 | 0.470 | 0.482 |
| Finetuned Llama | **0.796** | **0.738** | **0.718** | **0.670** |
| *Models with different PEs* | | | | |
| SPE | 0.67 | 0.59 | 0.56 | 0.55 |
| LPE | 0.67 | 0.61 | 0.57 | 0.56 |
| NoPE | 0.63 | 0.58 | 0.53 | 0.53 |
| RoPE | 0.70 | 0.58 | 0.54 | 0.52 |

Table 3: Effectiveness of axiomatic training for d-separation under two training paradigms: training a model from scratch and fine-tuning a pretrained Llama model.

of graphs with nodes from $[3, 6]$ with branching factor in range $[0.6, 0.8]$ and uses the same premises from the `TS2` training set (see Sec 5.1). Given a causal graph *premise*, we create the hypotheses as follows: First, select all pairs of nodes $(x_1, x_2)$ in the graph, then select the conditioning set $\mathcal{C}$ from the remaining sets of nodes of size up to 5 nodes. The ground truth labels denote whether $x_1$ and $x_2$ are d-separated given the conditioning set $\mathcal{C}$. From this exhaustive set of hypotheses, we randomly subsample 175k instances.

### 6.2. Evaluation

Unlike transitivity that primarily involves reasoning over linear chains, d-separation is more challenging due to its dependence on various structural patterns, including colliders, chains, and forks. Given the difficulty of the task, we evaluate two models: **1)** 67M model trained from scratch on the d-separation axiomatic data; **2)** Llama-3-8B model finetuned on the same axiomatic data. We consider two evaluation settings: longer sequences with random flipping (refer Table A7) and branching (refer Table 3).

Overall, the model trained with RoPE achieves the highest performance among all models trained from scratch, with the NoPE-based model ranking a close second across both complex graph structures. GPT-4, in contrast, struggles to outperform random baselines in both settings. For complex linear graphs, our RoPE-trained model significantly exceeds random baseline performance and clearly outperforms both GPT-4 and Llama-3-8B-Instruct. The axiomatically finetuned model performs comparably and maintains its effectiveness even as graph length increases, whereas the RoPE-trained model's performance declines for longer sequences. For branched causal graphs, although the RoPE-trained model initially performs well, its accuracy quickly drops toward the random baseline as complexity grows. In these more challenging scenarios, the Llama-finetuned model remains consistently strong and surpasses GPT-4 across all node sizes, thus highlighting the impact of axiomatic training in this setup.

## 7. Axiomatic finetuning for complex tasks

We now focus on causal reasoning benchmarks such as Corr2Cause and CLEAR and show how axiomatic finetuning can help improve causal reasoning of pretrained language models. We finetune Llama-3.1-8B-Instruct using supervised fine-tuning on our axiomatic training data (described in Sec. 4.1). Finetuning an existing model avoids any restriction on tokens and language style; the axiomatically finetuned model can be applied to any general text. We use Llama-3.1-8B-Instruct as the baseline to assess the improvement through axiomatic finetuning. We also include GPT-4 as a baseline, given its demonstrated strength in reasoning tasks.

### 7.1. Evaluation on CLEAR Benchmark

**Dataset and Evaluation Setup.** CLEAR (Chen et al., 2024) provides a comprehensive benchmark assessing language models' understanding of causal graphs through 20 diverse tasks, including d-separation. It has different question types like Multi-Choice and Yes/No type questions (Refer Fig. 1 for an example instance).

We test our model, fine-tuned on axiomatic d-separation instances, in a zero-shot setting on CLEAR's Yes/No (YN) and Multi-Choice (MC) questions. Note that there are multiple differences, including in structure and wording, between the finetuning data and the test benchmark. In addition, our finetuning data does not include any MC questions, only YN questions. Our goal is to check whether a axiomatically finetuned Llama model would generalize zero-shot to new kinds of causal questions.

**Results.** Table 5 shows the results. Our axiomatically finetuned Llama model provides a significant improvement in accuracy over the base model, for both YN and MC questions. Specifically, on MC questions, fine-tuning on axiomatic instances of d-separation yields a 17% improvement over the base model, even surpassing GPT-4. These strong zero-shot improvements across different question types and graph structures—notably outperforming larger models like GPT-4, indicate the effectiveness of axiomatic finetuning in enhancing causal reasoning.

### 7.2. Evaluation on Corr2Cause Dataset

**Dataset details.** Corr2Cause (Jin et al., 2024b) provides a more complex dataset to evaluate models on different causal tasks. Each data instance in the benchmark includes correlational relationships described in natural language for graphs with 3 to 6 nodes; the goal is to infer the truth value of a hypothesis. The hypothesis consists of six different kinds of graphical relationships between pairs of variables: Parent, Ancestor, Child, Descendant, Collider, and Confounder. This task is significantly harder than applying a

| Model | F1 | Prec | Rec | Acc |
|---|---|---|---|---|
| Llama-3-8b-Instruct | 0.11 | 0.15 | 0.08 | 0.77 |
| Llama-3-8b-Instruct finetuned Dsep (175k) | 0.23 | 0.16 | 0.46 | 0.54 |
| Llama-3-8b-Instruct finetuned Transitivity (175k) | **0.32** | 0.20 | 0.93 | 0.39 |
| GPT-4 (from (Jin et al., 2024a)) | 0.29 | 0.21 | 0.48 | 0.64 |

Table 4: **Evaluation on the Corr2Cause Task from (Jin et al., 2024a).** We finetune Llama-3-8B on our transitivity and d-separation datasets and evaluate on the Corr2Cause dataset in a zero-shot setting. We observe a significant increase in performance (over 20 p.p. improvement in F1 score compared to the base Llama model).

| CLEAR D-Separation task (YN) | |
|---|---|
| **Models** | **Accuracy** |
| Llama-3-8b-Instruct | 60.0 |
| Llama-3-8b-Instruct Finetuned | 70.0 |
| GPT-4 | 63.33 |
| **CLEAR D-Separation task (MC)** | |
| **Models** | **Accuracy** |
| Llama-3-8b-Instruct | 33.0 |
| Llama-3-8b-Instruct Finetuned | 50.0 |
| GPT-4 | 36.67 |

Table 5: **Evaluation on CLEAR dataset (Chen et al., 2024)** We finetune the LMs on our d-separation dataset and evaluate on the CLEAR dataset. We observe a significant increase in performance compared to the baseline, which highlights the efficacy of axiomatic training.

single axiom. First, one needs to infer the causal graph from the correlation statements. This requires knowledge of d-separation and Markov condition (Appendix E.1). Then, one needs to use the transitivity axiom to identify the direct and indirect relationships to identify the children, ancestors, colliders, and confounders in the causal graph.

It was found that popular LLMs fail to perform well on this benchmark. While finetuning on this dataset shows promise, the finetuned language models also fail in out-of-distribution settings generated by simply perturbing these questions or changing node names. As for the CLEAR benchmark, we will test our axiomatically finetuned models exactly in the out-of-distribution setting: the models have only been trained on synthetic axiomatic data (described in Section 4.1) and not seen any instance of Corr2Cause. Specifically, we finetune two models, one based on transitivity data and one based on d-separation data, given that the questions involves multiple axioms. Due to the high imbalance between labels in the dataset, we use F1 score to measure performance.

**Results.** Our results highlight Llama-3-8b-Instruct Base model's poor performance on this task, while axiomatic fine-tuning results in significant performance gains of around

20 p.p. (Table 4). Fine-tuning on both transitivity and d-separation improves performance. Notably, transitivity fine-tuning led to the largest gains, even surpassing GPT-4, likely due to its direct relevance in inferring graph relationships required for the hypothesis covered in the Corr2Cause benchmark (such as identifying child, ancestor, parent, descendant). The significant gains in F1 scores under an out-of-distribution setting demonstrate the effectiveness of axiomatic finetuning.

## 8. Discussion and Conclusion

We introduced a general framework, *axiomatic training*, to add axioms and simple rules of causality as inductive priors in ML models and showed its effectiveness in improving performance on downstream causal reasoning tasks. For applicability to real-world tasks, we also showed that it can be used to finetune pre-trained language models. Our study shows the importance of diversity in axiomatic data to enhance generalizability.

**Generalization to Logical Reasoning.** While our axiomatic training approach focuses on causal reasoning, it can be applied to any formal system such as deductive logical reasoning. Recent work (Saparov et al., 2023) highlights LLMs' deterioration in performance as reasoning depth increases. Given the similarity between our setup and such tasks, it would be interesting to explore if axiomatic training can improve deductive reasoning in language models. For example, transformers may be trained on logical axioms to improve language model's reasoning; Morishita et al. (2024) provide promising evidence in this direction.

**Verifiers for Causal Reasoning.** Another application of axiomatic training may be in building verifiers that operate over language models' output. For instance, our axiomatically finetuned models can be used to detect violations of causal rules such as transitivity and d-separation in a given text, and give inference-time feedback to the language model to improve its generation.

**Implications for Training Language Models.** Incorporating causal axiom demonstrations into pretraining or finetuning can boost reasoning in small language models, enabling models like Phi-3 to reach GPT-4-level accuracy on causal tasks. Papadimitriou & Jurafsky (2023) show that pretraining on synthetic formal languages introduces beneficial inductive biases. Similarly, pretraining on synthetic axiomatic data may enhance language models' reasoning abilities. Another promising direction is to use axiomatic training to enhance base language models before applying post-training with reinforcement learning (RL). Since RL-based alignment typically amplifies the capabilities of a base model, incorporating causal understanding through axiomatic training could provide a stronger foundation (Gandhi et al., 2025).

## Impact Statement

This paper presents work whose goal is to advance the field of Machine Learning, by proposing effective training strategies to improve causal reasoning capabilities of language models. Robust causal reasoning has clear upside: decision-support tools in medicine, public policy, and the social sciences may become more trustworthy under distribution shifts and easier for experts to scrutinize. We also recognize possible downsides, flawed or adversarially fine-tuned models could generate spurious causal claims that reinforce bias or justify harmful interventions. To mitigate these risks, we release code and synthetic training data openly, detail known limitations, and encourage independent replication and auditing. On balance, we believe that equipping language models with transparent, principled causal reasoning capabilities will yield net benefits for society while enabling informed oversight.

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

## Appendix

| Query Type (Train/ Eval) | Data Instance Example (Premise-Hypothesis-Label) | Structure Type | Network Size (number of nodes) |
|---|---|---|---|
| **Transitivity** | | | |
| Train | Mhb causes iqB. iqB causes G. Does G cause iqB?: No | Short Linear Sequence | 3-6 |
| Train | N5w causes s. 6D causes s. Does N5w cause s?: Yes | Short Sequence with Random Flipping | 3-6 |
| Eval | w3 causes ROv. w3 causes tQC. H causes ROv. H causes tQC. b causes ROv. b causes w3. b causes H. Does tQC cause ROv?: No | Branching | 5,8,10,12 |
| Eval | LKk causes 5Ov. Kk causes L0. L0 causes KWO. 5Ov causes c. Does KWO cause L0?: No | Shuffled Sequences | 3-9 |
| Eval | FDAH26mV7 causes 7tzaIHjlY. 7tzaIHjlY causes 0kspcX95Im. 0kspcX95Im causes 7rhFSlx2o9. 7rhFSlx2o9 causes 1PlG5LHVqp. Does FDAH26mV7 cause 7tzaIHjlY?: Yes | Sequences with Longer Node Names | 3-9 |
| Eval | r causes rZ. rZ causes L. L causes bUx. bUx causes Pbr. Pbr causes 1w. 1w causes c3. c3 causes yBQ. yBQ causes yK. yK causes w. w causes P. P causes kH. kH causes 1u. 1u causes jV7. jV7 causes i. Does r cause rZ?: Yes | Long Linear Sequences | 7-15 |
| Eval | rU6 causes eF. eF causes ivC. 3R causes ivC. 3R causes A8. 2 causes A8. 2 causes i. i causes a03. y causes a03. b causes y. b causes h. h causes yN. ic0 causes yN. ic0 causes Hd. Hd causes U. Does rU6 cause eF?: Yes | Long Sequences with Random Flipping | 7-15 |

Table A1: Table with examples of transitivity data instances of different causal structural networks used for training and evaluating models.

## A. Transitivity Axioms

**Length Generalization:** Table A4 shows the accuracy of different models when evaluated on larger causal chains that were not seen during training. Among the baseline pre-trained LMs, GPT-4 obtains the highest accuracy on both sequential and randomly flipped chains for the multi-shot setting. It is remarkable that our TS2 (NoPE) model obtains competitive performance to the trillion-scale GPT-4 model. In particular, for chains of size 7-12, TS2 (NoPE) obtains higher or comparable accuracy than GPT-4 on both sequential and randomly flipped chains. Similar trends are observed for chains of size 7-13 when compared to GPT-4 in the zero-shot setting. Our model's accuracy decreases for chains of length 14-15 (0.85 for sequential chains and 0.78 for randomly flipped chains) but is still significantly higher than that of LMs like Gemini-Pro and Phi-3. Although in-context examples in *multi-shot* setting improve the performance of baseline LLMs, TS2 (NoPE) still outperforms both Gemini Pro and Phi-3 in the multi-shot setting. Note that a random prediction would yield a 50%

| Query Type (Train/ Eval) | Data Instance Example (Premise-Hypothesis-Label) | Structure Type | Network Size (number of nodes) |
|---|---|---|---|
| **D-Separation** | | | |
| Train | 1c1 causes kT. kT causes t4d. t4d causes zW. zW causes Z4P. Z4P causes pij. Are zW and pij d-separated given {t4d, kT, Z4P}?: Yes | Short Linear Sequence with Non Empty Conditioning Set | 6 |
| Train | nL causes A. A causes xx. xx causes 5Cg. Are xx and nL d-separated?: No | Short Linear Sequence with Empty Conditioning Set | 4 |
| Train | ZWn causes P9. u causes P9. B causes u. NS causes B. Are P9 and u d-separated given {B}?: No | Short Sequence with Empty Conditioning Set | 5 |
| Train | ZWn causes P9. u causes P9. B causes u. NS causes B. Are P9 and u d-separated given {B, ZWn}?: No | Short Sequence with Empty Conditioning Set | 5 |
| Eval | FZg causes l. FZg causes Y. Y causes vEU. a causes vEU. f5 causes a. f5 causes R. R causes O. 2WJ causes O. 2WJ causes TDA. TDA causes 9d. Are 2WJ and 9d d-separated given {FZg}? | Long Sequence with Random Flipping | 10 |
| Eval | t causes a. t causes OP. t causes wT. faG causes t. faG causes Z. pK causes OP. pK causes 0K3. pK causes yUa. 0K3 causes a. 0K3 causes OP. 0K3 causes wT. Z causes yUa. rY6 causes faG. rY6 causes wT. Are t and yUa d-separated given {faG}?: Yes | Branching | 12 |

Table A2: Table with examples of d-separation data instances of different causal structural networks used for training and evaluating models. We have instances with null and non empty conditioning set for d-separation in our train and evaluation set.

accuracy, indicating that the axiomatically-trained `TS2 (NoPE)` model can generalize its reasoning to causal chains much longer than 6 even though it was trained only on chains up to length 6.

**Node Name Shift:** For models trained on `TS2` dataset, we also evaluate generalization to changes in variable names (Figure 2). We find that `TS2 (NoPE)` is robust to node name changes and retains its high accuracy as new, longer names are introduced. It also retains its generalizability to longer sequences with new node names, performing similarly to GPT-4.

**Summary:** Across all evaluation setups, our axiomatically trained model `TS2 (NoPE)` performs significantly better than random baselines even as chain lengths are increased beyond its training data. In particular, even though our model was not trained on fully reversed chains, it performs at par with the significantly larger GPT-4 model, while easily outperforming other billion scale models even under multi-shot settings. For other tasks, it often outperforms or matches the accuracy of billion-scale models like Gemini Pro and Phi-3. These results indicate that a model trained axiomatically can learn to reason about more complex causal structures from demonstrations of simple causal sequences.

| Model | 3 | 4 | 5 | 6 |
|---|---|---|---|---|
| **Baselines** | | | | |
| **Zero Shot** | | | | |
| GPT-4 | 0.97 | 0.99 | 0.98 | 0.92 |
| Gemini Pro | 0.61 | 0.59 | 0.66 | 0.62 |
| Phi-3 | 0.80 | 0.69 | 0.73 | 0.69 |
| **Multi Shot** | | | | |
| GPT-4 | **1.00** | **1.00** | **1.00** | **0.99** |
| Gemini Pro | 0.95 | 0.87 | 0.77 | 0.71 |
| Phi-3 | 0.93 | 0.89 | 0.75 | 0.75 |
| **Axiomatic Training** | | | | |
| TS1 w NoPE | 0.99 | 0.97 | 0.90 | 0.91 |
| TS1 w LPE | 0.99 | 0.98 | 0.95 | 0.93 |
| TS1 w SPE | **1.00** | 0.98 | 0.95 | 0.96 |
| TS1 w RoPE | 0.97 | 0.97 | 0.96 | 0.98 |
| TS2 w NoPE | 0.98 | 0.96 | 0.90 | 0.91 |
| TS2 w LPE | 0.99 | 0.97 | 0.92 | 0.96 |
| TS2 w SPE | 0.99 | 0.97 | 0.93 | 0.94 |
| TS2 w RoPE | 0.99 | 0.98 | 0.97 | 0.98 |
| OCC w NoPE | 0.41 | 0.24 | 0.18 | 0.13 |
| OCC w RoPE | 0.59 | 0.26 | 0.22 | 0.20 |

Table A3: Following (Berglund et al., 2024), we evaluate models on inferring cause-and-effect from fully reversed sequences absent in training data. Models trained on OCC perform worse, highlighting the importance of edge-level perturbations for generalization. Accuracy metric is reported, with random baseline = 0.5. Best performance is bolded, while second best is underlined.

### A.1. Additional Results: Role of Data Diversity and Positional Encoding

**Importance of Data Perturbations.** We find that diversity of the sequences in train data plays an important role. Model trained on only causal chains (OCC) generalize to longer chains (Table A4) but not to other DAG structures (see Figure 3 for edge flip, Table 2 for branching). Models trained on TS1 or TS2 generalize across all scenarios, including random flip, order permutations, and branching; thus highlighting the impact of incorporating variability at the edge level through random flipping. However, across tasks, we find that TS2 yields higher accuracy than TS1, even as TS1 has more variations due to random flipping. This suggests that while perturbations aid structural generalization, excessive perturbations can hinder it (in particular, random flipping may decrease the length of available causal paths during training).

**Role of Positional Encodings.** When comparing models based on positional encoding, we find that models without positional encoding generalize well to both longer chains (up to length 15) and unseen complex graph structures, despite being trained only on linear chains with 3-6 nodes. Models with SPE and LPE perform well on longer chains but struggle with longer node names, even in smaller graphs (Figure 2), highlighting their sensitivity to minor perturbations. SPE also underperforms in branching and order-based settings like shuffling and reversal. Learnable PE works up to 9-length chains but drops afterward. Overall, our results extend earlier work on the utility of NoPE (Kazemnejad et al., 2023; Haviv et al., 2022) to the task of understanding causal sequences and generalizing to both longer length and complex structure at test time. Interestingly, all PEs perform well in randomly flipped sequences, likely due to the short effective path lengths caused by the 0.5 probability of forward-directed edges.

## B. Zero Shot Setting

To evaluate the baseline models, we follow a simple zero-shot prompting strategy. For each tuple, we provide the natural language expression of the causal graph *(Premise)* followed by the question *(Hypothesis)* and prompt the LM to answer it in either 'Yes' or 'No' *(Label)*. Here is an example prompt: *"EX causes T. T causes 9. 9 causes W. W causes 7. 7 causes M. M causes a. Does EX cause T? Answer in 'Yes' or 'No' only"*. See Table **??** contains examples of prompts used.

## C. Multi-Shot Prompt

### C.1. Cause-Effect Inference Task

Chain lengths of the in context examples ranged from 3 to 6 to maintains consistency with the training and testing paradigm used for our 67-million-parameter model.

The following multi-shot prompt was used to evaluate the baselines and models across different test sets, assessing their generalization based on length, order, and branching.

*Following the given examples answer the question regarding causal relationship between two variables: '5e0 causes vAf. vAf causes VO. Does vAf cause VO?: Yes'*
*'5e0 causes vAf. vAf causes VO. Does vAf cause 5e0?: No'*
*'e0F causes Z. Z causes 0U. 0U causes mR. mR causes 1L. Does mR cause 1L?: Yes'*
*'e0F causes Z. Z causes 0U. 0U causes mR. mR causes 1L. Does Z cause e0F?: No'*
*'b causes K. K causes qPv. 5 causes qPv. Does b cause qPv?: Yes'*
*'b causes K. K causes qPv. 5 causes qPv. Does b cause 5?: No'*
*'Mhb causes t0a. 6Eh causes Mhb. NS causes 6Eh. n causes NS. n causes xu. Does xu cause 6Eh?: No'*
*'Mhb causes t0a. 6Eh causes Mhb. NS causes 6Eh. n causes NS. n causes xu. Does n cause NS?: Yes'*

# D. Results and Analysis

| Model | 7 | | 8 | | 9 | | 10 | | 11 | | 12 | | 13 | | 14 | | 15 | |
|---|---|---|---|---|---|---|---|---|---|---|---|---|---|---|---|---|---|---|
| | FS | RF | FS | RF | FS | RF | FS | RF | FS | RF | FS | RF | FS | RF | FS | RF | FS | RF |
| **Baselines** | | | | | | | | | | | | | | | | | | |
| **Single Shot** | | | | | | | | | | | | | | | | | | |
| GPT-4 | 0.95 | 0.98 | 0.97 | 0.93 | 0.87 | 0.94 | 0.91 | 0.87 | **0.90** | 0.95 | 0.92 | 0.92 | 0.85 | 0.93 | **0.93** | 0.93 | **0.89** | 0.86 |
| Gem-Pro | 0.63 | 0.73 | 0.69 | 0.74 | 0.64 | 0.75 | 0.65 | 0.81 | 0.72 | 0.78 | 0.60 | 0.80 | 0.59 | 0.68 | 0.67 | 0.64 | 0.61 | 0.66 |
| Phi-3 | 0.81 | 0.85 | 0.96 | 0.85 | 0.85 | 0.85 | 0.87 | 0.89 | **0.90** | 0.86 | 0.84 | 0.85 | 0.91 | 0.84 | 0.90 | 0.80 | 0.78 | 0.85 |
| **Multi Shot** | | | | | | | | | | | | | | | | | | |
| GPT-4 | 0.97 | **0.99** | 0.93 | **0.99** | 0.92 | **0.96** | 0.88 | **0.94** | 0.89 | **0.97** | 0.89 | **0.93** | 0.88 | **0.95** | 0.93 | **0.94** | 0.86 | **0.94** |
| Gem-Pro | 0.80 | 0.82 | 0.81 | 0.79 | 0.78 | 0.81 | 0.67 | 0.79 | 0.73 | 0.82 | 0.74 | 0.83 | 0.67 | 0.78 | 0.72 | 0.78 | 0.68 | 0.78 |
| Phi-3 | 0.83 | 0.92 | 0.89 | 0.88 | 0.75 | 0.86 | 0.66 | 0.87 | 0.80 | 0.90 | 0.80 | 0.85 | 0.79 | 0.82 | 0.71 | 0.81 | 0.72 | 0.82 |
| **Axiomatic Training** | | | | | | | | | | | | | | | | | | |
| TS1 w NoPE | 0.99 | 0.96 | 0.97 | 0.95 | 0.86 | 0.92 | 0.78 | 0.87 | 0.77 | 0.90 | 0.76 | 0.82 | 0.77 | 0.82 | 0.75 | 0.83 | 0.70 | 0.76 |
| TS1 w LPE | 0.98 | 0.96 | 0.89 | 0.94 | 0.81 | 0.91 | 0.61 | 0.86 | 0.64 | 0.87 | 0.64 | 0.79 | 0.60 | 0.80 | 0.59 | 0.81 | 0.57 | 0.73 |
| TS1 w SPE | 0.99 | 0.91 | 0.88 | 0.92 | 0.73 | 0.77 | 0.62 | 0.69 | 0.63 | 0.65 | 0.69 | 0.60 | 0.62 | 0.62 | 0.59 | 0.58 | 0.63 | 0.58 |
| TS1 w RoPE | 0.99 | 0.96 | 0.97 | 0.95 | 0.89 | 0.90 | 0.82 | 0.84 | 0.81 | 0.84 | 0.86 | 0.76 | 0.76 | 0.81 | 0.82 | 0.70 | 0.78 | 0.75 |
| TS2 w NoPE | 0.98 | 0.93 | 0.93 | 0.92 | 0.82 | 0.88 | 0.74 | 0.84 | 0.70 | 0.85 | 0.70 | 0.80 | 0.71 | 0.76 | 0.71 | 0.77 | 0.66 | 0.73 |
| TS2 w LPE | 0.99 | 0.95 | 0.96 | 0.94 | 0.86 | 0.90 | 0.72 | 0.86 | 0.69 | 0.85 | 0.80 | 0.78 | 0.73 | 0.78 | 0.75 | 0.80 | 0.68 | 0.77 |
| TS2 w SPE | 0.97 | 0.92 | 0.91 | 0.92 | 0.76 | 0.85 | 0.58 | 0.72 | 0.60 | 0.66 | 0.61 | 0.56 | 0.60 | 0.56 | 0.58 | 0.56 | 0.56 | 0.59 |
| TS2 w RoPE | 0.99 | 0.97 | 0.98 | 0.96 | 0.90 | 0.89 | 0.85 | 0.87 | 0.84 | 0.82 | 0.87 | 0.74 | 0.78 | 0.80 | 0.86 | 0.69 | 0.78 | 0.71 |
| OCC w NoPE | 0.99 | 0.61 | 0.98 | 0.62 | 0.89 | 0.62 | 0.90 | 0.57 | 0.90 | 0.57 | 0.93 | 0.52 | 0.87 | 0.55 | 0.93 | 0.50 | 0.87 | 0.53 |
| OCC w RoPE | 0.96 | 0.65 | 0.98 | 0.71 | 0.84 | 0.68 | 0.84 | 0.64 | 0.80 | 0.65 | 0.88 | 0.56 | 0.76 | 0.60 | 0.84 | 0.60 | 0.79 | 0.55 |

Table A4: **Accuracy of different models on Transitivity axioms.** In this table, we show the accuracy of different models on the transitivity axioms. The rows shows different models considered for comparison. The top rows denote the performance of baseline models with different prompting strategies i.e. single shot and multi-shot prompt (see Sec 4.5 for details). The models listed after axiomatic training shows the performance of transformer models trained from scratch on our axiomatic dataset. TS1 and TS2 denote pretraining data setups 1 and 2 as described in Sec 5.1 and different modifiers are: SPE: Sinusoidal Positional Encoding (PE), LPE: Learnable PE, w/o PE: No PE, RoPE: Rotary Position Embedding. For axiomatic training, the model remains the same across all setups (67 Million parameters based). The training dataset contain graphs of size 3-6 however the models are tested on graphs of size 7-15 (as mentioned in different columns). FS denotes the graphs that only contain chains that are oriented in forward direction and RF contains the graphs that also includes random flipping (see Sec 4.1 for details) same as training set.

| Model | 3 | 4 | 5 | 6 | 7 | 8 | 9 |
|---|---|---|---|---|---|---|---|
| **Baselines** | | | | | | | |
| **Single Shot** | | | | | | | |
| GPT-4 | **1.00** | **1.00** | **1.00** | **1.00** | **1.00** | **1.00** | **1.00** |
| Gemini Pro | 0.96 | 0.94 | 0.86 | 0.81 | 0.76 | 0.73 | 0.71 |
| Phi-3 | 0.99 | 0.98 | 0.95 | 0.94 | 0.96 | 0.95 | 0.93 |
| **Multi Shot** | | | | | | | |
| GPT-4 | **1.00** | **1.00** | 0.98 | 0.98 | 0.98 | 0.98 | 0.97 |
| Gemini Pro | **1.00** | **1.00** | 0.91 | 0.90 | 0.86 | 0.88 | 0.84 |
| Phi-3 | 0.93 | 0.89 | 0.89 | 0.84 | 0.82 | 0.77 | 0.79 |
| **Axiomatic Training** | | | | | | | |
| TS1 w NoPE | **1.00** | **1.00** | **1.00** | 0.99 | 0.98 | 0.92 | 0.88 |
| TS1 w LPE | **1.00** | **1.00** | 0.99 | 0.97 | 0.92 | 0.83 | 0.74 |
| TS1 w SPE | 0.76 | 0.61 | 0.58 | 0.57 | 0.54 | 0.50 | 0.54 |
| TS1 w RoPE | 0.65 | 0.56 | 0.56 | 0.49 | 0.45 | 0.49 | 0.50 |
| TS2 w NoPE | **1.00** | 0.99 | 0.92 | 0.84 | 0.76 | 0.71 | 0.69 |
| TS2 w LPE | **1.00** | 0.99 | 0.96 | 0.90 | 0.86 | 0.76 | 0.74 |
| TS2 w SPE | 0.82 | 0.66 | 0.60 | 0.58 | 0.57 | 0.55 | 0.53 |
| TS2 w RoPE | 0.51 | 0.48 | 0.48 | 0.50 | 0.46 | 0.46 | 0.48 |
| OCC w NoPE | **1.00** | 0.99 | 0.98 | 0.96 | 0.96 | 0.91 | 0.93 |
| OCC w RoPE | 0.90 | 0.77 | 0.67 | 0.64 | 0.65 | 0.59 | 0.62 |

Table A5: Results on node name length generalization. TS1 and TS2 denote Training Data setup 1 and 2 from Section 4 **??**. OCC is the third data setup comprising of sequential causal chains. SPE: Sinusoidal PE, LPE: Learnable PE, w/o PE: No PE, RoPE: Rotary Position Embedding. Model remains the same across all setups (67 Million parameter based). For longer node names, NoPE performs best on sequential linear setup. Accuracy metric is used.

| Model | 5 | | 8 | | 10 | | 12 | |
|---|---|---|---|---|---|---|---|---|
| | BF=2 | BF=1.4 | BF=2 | BF=1.4 | BF=2 | BF=1.4 | BF=2 | BF=1.4 |
| **Baselines** | | | | | | | | |
| *Zero shot* | | | | | | | | |
| GPT-4 | 0.98 | 0.95 | 0.91 | 0.90 | 0.84 | 0.88 | 0.82 | 0.86 |
| Gemini Pro | 0.77 | 0.74 | 0.72 | 0.76 | 0.71 | 0.73 | 0.73 | 0.71 |
| Phi-3 | 0.87 | 0.83 | 0.82 | 0.79 | 0.77 | 0.77 | 0.75 | 0.80 |
| *Multi shot* | | | | | | | | |
| GPT-4 | **0.99** | **0.97** | **0.94** | **0.93** | **0.90** | **0.94** | **0.89** | **0.93** |
| Gemini Pro | 0.81 | 0.76 | 0.77 | 0.79 | 0.75 | 0.77 | 0.78 | 0.79 |
| Phi-3 | 0.77 | 0.78 | 0.79 | 0.82 | 0.78 | 0.79 | 0.80 | 0.79 |
| **Axiomatic Training** | | | | | | | | |
| OCC w RoPE | 0.74 | 0.72 | 0.70 | 0.68 | 0.66 | 0.66 | 0.65 | 0.62 |
| TS1 w LPE | 0.76 | 0.82 | 0.70 | 0.72 | 0.67 | 0.69 | 0.63 | 0.68 |
| TS1 w SPE | 0.65 | 0.78 | 0.57 | 0.61 | 0.55 | 0.59 | 0.53 | 0.57 |
| TS1 w NoPE | 0.78 | 0.82 | 0.70 | 0.74 | 0.67 | 0.69 | 0.62 | 0.66 |
| TS1 w RoPE | 0.81 | 0.86 | 0.75 | 0.79 | 0.73 | 0.74 | 0.69 | 0.71 |
| TS2 w LPE | 0.73 | 0.80 | 0.68 | 0.72 | 0.65 | 0.67 | 0.61 | 0.64 |
| TS2 w SPE | 0.65 | 0.79 | 0.53 | 0.59 | 0.52 | 0.54 | 0.52 | 0.52 |
| TS2 w NoPE | 0.75 | 0.82 | 0.68 | 0.73 | 0.67 | 0.68 | 0.62 | 0.64 |
| TS2 w RoPE | 0.81 | 0.88 | 0.74 | 0.79 | 0.70 | 0.72 | 0.68 | 0.68 |

Table A6: **Evaluation with variation in branching factor.** Accuracy of axiomatically trained models with LM baselines on the causal graphs with higher branching factor than that in training. See Sec 5.2 for details.

# E. Preliminaries and Notations

**Causal Models** Let $\mathcal{M} = (\boldsymbol{X}, \boldsymbol{U}, \mathcal{F})$ be a causal model defined over a set of endogenous variables $\boldsymbol{X}$, exogenous variables $\boldsymbol{U}$ and the causal relationship between then defined by set of structural equations $\mathcal{F}$ (Galles & Pearl, 1997). Let $\mathcal{G}$ be the causal graph associated with the causal model $\mathcal{M}$ where the nodes $\boldsymbol{V}$ in $\mathcal{G}$ correspond to the variables in $\mathcal{M}$ and an edge $V_i \rightarrow V_j$ between any two nodes $V_i, V_j$ denote the causal relationship between them. The causal relationship of node $X_i$ is characterized by the functional relationship $f_i \in \mathcal{F}$ s.t., $x_i = f_i(\boldsymbol{pa}_i, \boldsymbol{u}_i)$. Here $\boldsymbol{pa}_i$ are the parent of the node $X_i$ is the corresponding causal graph $\mathcal{G}$ and $\boldsymbol{u_i} \subseteq \boldsymbol{U}$ are set of exogenous variables influencing the exogenous variable $X_i$. In our work, we assume that there are no hidden confounders so we have one exogenous variable corresponding to every endogenous variable i.e. $\boldsymbol{u}_i = u_i$. Each exogenous variable has an associated probability distribution which quantifies the uncertainty in the system i.e. $u_i \sim \mathbb{P}(u_i)$. Thus the joint distribution of the exogenous variable is given by $\mathbb{P}(\boldsymbol{U})$. Since any endogenous variable is a deterministic function of other endogenous and exogenous variables the probability distribution corresponding to the endogenous variable is the push-forward of the exogenous variable i.e $\mathbb{P}(\boldsymbol{X}) \triangleq \mathbb{P}^{\#}(\boldsymbol{U})$.

## E.1. Definitions

Following the formal definitions provided by (Jin et al., 2024a), we explain the following terminologies:

**Markov Property** In a directed acyclic graph (DAG) $G$, the Markov property asserts that each node $X_i$ is conditionally independent of its non-descendants given its parents. This can be written as $X_i \perp\!\!\!\perp \text{NonDe}(X_i) \,|\, \text{Pa}(X_i)$, where $\text{NonDe}(X_i)$ represents the set of non-descendants of $X_i$, excluding the node itself, and $\text{Pa}(X_i)$ denotes its parents. Leveraging the Markov property, the joint distribution over all the nodes can be factorized as:

$$P(X_1, \ldots, X_N) = \prod_{i=1}^{N} P(X_i \,|\, \text{Pa}(X_i)).$$

**Markov Equivalence Class** Two directed acyclic graphs (DAGs) are considered Markov equivalent if they induce the same joint distribution $P(X)$. The collection of DAGs that are Markov equivalent is referred to as a Markov equivalence class (MEC). Causal graphs within the same MEC can be easily recognized as they share the same skeleton (i.e., undirected edges) and V-structures (i.e., configurations of the form $A \rightarrow B \leftarrow C$, where $A$ and $C$ are not directly connected).

## F. Positional Encodings and their Role in Generalization

Positional Encoding (PE) play a crucial role of providing information about the absolute and relative position of tokens in a sequence (Vaswani et al., 2017). (Vaswani et al., 2017) propose an absolute positional encoding strategy using periodic functions (e.g., sinusoidal or cosine) to initialize these encodings. Absolute positional encoding provides definite values for all positions across any sequence length. However, studies (Ontañón et al., 2022; Csordás et al., 2021) show absolute positional encoding fails in length generalization tasks for transformers. In the learnable APE variant (Radford et al., 2018), each positional embedding is randomly initialized and trained with the model. This approach falters with sequences longer than those seen in training, as the new positional embeddings remain untrained and randomized. Interestingly, recent findings (Kazemnejad et al., 2023; Haviv et al., 2022) indicate that removal of PEs in auto-regressive models can improve model's length generalization capabilities, wherein the attention mechanism during auto-regressive decoding is sufficient to encode positional information. We also experiment with Rotary Position Encodings (Su et al., 2024), which have shown superior length generalization. We use $\theta = 10000.0$ for the base period of RoPE embeddings.

## G. Formalising training and Evaluation Setup

Let $f_{dim}$ represent the maximum value for a given perturbation dimension $dim$, along which we construct train and evaluation sets for our axiomatic framework. For each dimension, we choose a threshold $\tau_{dim} \in L$, such that $f_{dim} < \tau_{dim}$ forms our training set and $f_{dim} \geq \tau_{dim}$ forms the evaluation set. So, $f_{dim} \in \{f_{len}, f_{branch}, f_{nodelen}, f_{revfactor}, f_{shuffle}\}$ where:

- $f_{len} = \max_{\forall i}(len(V_i))$, gives the maximum number of nodes across all causal sequences. $\tau_{len}$ for length is set at 6, with $f_{len} \in [3, 6]$.

- $f_{branch} = \max_{\forall i}(|X_i|/|V_i|)$ gives the maximum branching factor in a dataset, with $\tau_{branch} = 0.8$ (for 6 node linear sequences). For sequences in the train set, the branching factor ranges from 0.6 to 0.8 for 3 to 6 length sequences.

- Let $l_{i,j}$ be the length of the name of the node $X_{i,j}$, then $l_{i,j} = (len(X_{i,j}))$. Therefore, the maximum length of node names across all nodes in all causal sequences can be represented as: $f_{nodenamelen} = \max_{1 \leq i \leq n, 1 \leq j \leq m} l_{i,j}$. We set $\tau_{nodelen}$ for train set as 3, with $f_{nodelen} \in [1, 3]$.

- Given any causal sequence $X_i$ and a function $N$, where $N(X_{i,j}, X_{i,j+1})$ returns natural language representation of a directed edge between $j$ and $j + 1$ node in the causal chain $X_i$. $f_{shuffle} = \cap_{\forall i,j}\text{Perm}(N(X_{i,j}, X_{i,j+1}))$, where $N(X_{i,j}, X_{i,j+1})$ represents deviation from original sequential order of natural language sentences to represent $X_i$.

- Given a causal sequence $X_i$ and let $R(X_i, f_{revfactor})$ be an operation on the causal chain that flips the direction of every edge in the sequence with probability $f_{revfactor}$. In the training set, there is a directed edge between every sequential pair of nodes $X_{i,j}, X_{i,j+1}$ with $f_{revfactor} = 0$ (for linear sequence, $X_{i,j} \to X_{i,j+1}$) or 0.5 (for sequence with random flipping, $X_{i,j} \to X_{i,j+1}$ or $X_{i,j} \leftarrow X_{i,j+1}$) In the evaluation set $f_{revfactor} = 1$ i.e., all sequences for reversal evaluation setup are completely reversed unlike in train set where no sequence is present where all edges are completely reversed.

## H. Results of Dsep on CLEAR dataset

| Models | 7 | 8 | 9 | 10 | 11 | 12 | 13 | 14 |
|---|---|---|---|---|---|---|---|---|
| GPT-4 | 0.57 | 0.64 | 0.52 | 0.50 | 0.53 | 0.52 | 0.51 | 0.50 |
| Llama-3-8b-Instruct | 0.51 | 0.50 | 0.54 | 0.51 | 0.48 | 0.51 | 0.53 | 0.51 |
| Llama-3-8b-Instruct-Finetuned | 0.952 | 0.948 | 0.954 | 0.850 | 0.87 | 0.88 | 0.73 | 0.66 |
| *Models with different PEs trained from scratch* | | | | | | | | |
| SPE | 0.93 | 0.95 | 0.97 | 0.95 | 0.71 | 0.61 | 0.80 | 0.44 |
| LPE | 0.96 | 0.93 | 0.99 | 0.92 | 0.68 | 0.71 | 0.62 | 0.47 |
| NoPE | 0.89 | 0.93 | 0.85 | 0.94 | 0.68 | 0.65 | 0.60 | 0.5 |
| RoPE | 0.96 | 0.91 | 0.96 | 0.92 | 0.70 | 0.69 | 0.54 | 0.58 |

Table A7: Performance on DSEP of longer chains with random flipping

## I. Examples of instances from each benchmark: CLEAR and Corr2Cause

> **Premise:** *Given a DAG (directed acyclic graph) with nodes C, Z, P, V and directed edges C->V, P->V, C->Z, Z->P, Z->V.*
> **Hypothesis:** "Which of the following nodesets can d-separate node C and node P?
> A. `{'Z', 'V'}`
> B. `{'V'}`
> C. `{'Z'}`
> D. `set()`
> **Answer:** *C*

Figure A1: Example instance of Multiple Choice (MC) question type from (Chen et al., 2024) dataset describing d-separation rule problem defined with a different hypothesis type and semantic structure then the one our models are finetuned on.

> **Premise:** *Suppose there is a closed system of 4 variables, A, B, C and D. All the statistical relations among these 4 variables are as follows: A correlates with B. A correlates with C. A correlates with D. B correlates with C. B correlates with D. C correlates with D. However, B and D are independent given A. B and D are independent given A and C. C and D are independent given A. C and D are independent given A and B.*
> **Hypothesis:** There exists at least one collider (i.e., common effect) of A and B.
> **Label:** *No.*

Figure A2: Example instance from the Corr2Cause dataset, where the model must infer the presence of a collider between variables given only correlational and conditional independence statements.

## J. Implementation Details

We used a learning rate of 1e-4 with linear scheduling and 3% warmup ratio, training for 4102 max steps on axiomatic instance samples with sequences of maximum length 4096 tokens. We employed mixed precision (bfloat16) training with flash attention for efficiency. After training, the LoRA weights were merged with the base model for inference. We used Huggingface (wol, 2020) for implementation. The fine-tuning used LoRA with rank 64, alpha 16, and dropout 0.1. Training was performed on 3 GPUs using DeepSpeed Stage 3 with a total batch size of 128 (16 samples per GPU with gradient accumulation). 1 A40 and 1 A100 GPUs were used for training the transformer model from scratch for all Positional encodings based experiments.

