# OpenReview forum: "Teaching Transformers Causal Reasoning through Axiomatic Training"
_ICML.cc/2025/Conference — ICML 2025 poster_

### Official Review · Reviewer_HBJz · 2025-02-19

**Overall Recommendation:** 4

**Summary:**

This paper proposes axiomatic training, which leverages synthetic data to train small models from scratch. The authors observed that their approach enables models to generalize from small-node to large-node causal structures when evaluated on transitivity axioms and d-separation rules. Moreover, fine-tuning Llama-7B with axiomatic training improved performance on the CLEAR and Corr2Cause benchmarks, surpassing the performance of larger models like GPT-4.

**Claims And Evidence:**

The authors argue that axiomatic pretraining on synthetic data helps models better understand causality, and that this capability generalizes effectively to other tasks. Improvements in two experimental settings support the claim, demonstrating the versatility of this approach.

**Essential References Not Discussed:**

https://arxiv.org/pdf/2405.15071, discusses the generalization capabilities of large models on synthetic dataset, could be relevant to this work. It may provide insights into the sources of generalization.

**Experimental Designs Or Analyses:**

* CLEAR dataset: Accuracy on the d-separation task (binary classification) improved from 30% to 70%, while multiple-choice accuracy improved from 33% to 50%.
* Corr2Cause dataset: F1 scores improved by 20%, outperforming GPT-4.
* For small models, the authors explored different positional embeddings and training strategies, providing insights for future pretraining approaches.

**Methods And Evaluation Criteria:**

The authors constructed a synthetic NLI dataset to pretrain models. They trained a 67M parameter small model from scratch and fine-tuned Llama-7B. The evaluation was conducted on the training task as well as on CLEAR and Corr2Cause datasets. These benchmarks are reliable indicators of causal reasoning performance.

**Other Comments Or Suggestions:**

Line 60: The authors left a comment that violates the anonymity principle.
Line 138: A formula exceeds the page margin and needs to be adjusted.

**Other Strengths And Weaknesses:**

See additional comments.

**Questions For Authors:**

* I have some concerns regarding the performance of the small models. Although the authors explored different training strategies and positional embeddings, the results appear to fall short of the prompting-based performance of existing large models.
* Could the authors clarify whether pretraining Llama-7B with this approach consistently outperforms prompting-based methods for large models in MultiEvalSLR Dataset?

**Relation To Broader Scientific Literature:**

This is an intriguing study. As natural datasets become increasingly scarce, the use of synthetic data as a supplement for training is an exciting direction for future research.

**Theoretical Claims:**

NA

---

> ### Author Rebuttal · Authors · 2025-04-01
>
> We thank the reviewer for appreciating the contributions of the work. We answer specific questions below.
>
> >**1. I have some concerns regarding the performance of the small models. Although the authors explored different training strategies and positional embeddings, the results appear to fall short of the prompting-based performance of existing large models.**
>
> **Response**:  While GPT-4 performs well on transitivity-based tasks, its accuracy drops to near-random on complex causal rules like d-separation (see Tables A6, A7). Despite its scale, prompting-based GPT-4 model struggles with d-separation, a key rule for inferring conditional independence in causal inference. In contrast, our 67M-parameter model achieves higher accuracy on d-separation, for complex graphs unseen during its training. This shows the potential of axiomatic training for causal tasks. Below we provide further practical evidence comparing prompting-based GPT-4 to an axiomatically finetuned Llama model.
>
> Finetuning Llama-3-8B-Instruct with axiomatic training further improves performance on complex graphs and the CLEAR benchmark (Table 3). On CLEAR’s multiple-choice task, the finetuned model achieves 50% accuracy, outperforming GPT-4’s 36.67% (with a random baseline of around 25%). The CLEAR benchmark has evaluation instances for d-separation that are semantically different from instances in the axiomatic training data. Also, CLEAR benchmark has different types of hypothesis (multiple choice question answer, yes/no type questions), whereas axiomatic instances only contained yes/no hypothesis instances, thus showing potential of the model to generalize for diverse problems beyond simple binary classification.
>
> Lastly, Table 4 shows results on the Corr2Cause benchmark, that evaluates the model's reasoning capability to infer causal relationships from correlational statements, shows the potential of axiomatic training on real world problems which require application of different causality rules (d-separation, transitivity, markov property, markov equivalence class). A Llama-3 model finetuned on axiomatic data obtains 0.37 F1 score and outperforms GPT-4 which struggles to perform well on the task (with 0.29 F1 score).
>
> >**2. Could the authors clarify whether pretraining Llama-7B with this approach consistently outperforms prompting-based methods for large models in MultiEvalSLR Dataset?**
>
> Below we present the results of applying the Llama-3-8b-Instruct model (finetuned with transitivity axiom data) to the MultiEvalSLR task.  We find that the Llama model performs at par with large prompt-based models like GPT-4.
>
> MultiEvalSLR is a complex evaluation set where each input premise contains different types of complexities compared to sequential causal chain: shuffling of premise, chains with randomly flipped edges, and longer chain lengths than the training instances. We train the Llama model on an axiomatic training dataset (same as in paper) that does not include any instances with shuffled premise. Up until length 6, its accuracy is better than GPT-4. However, the GPT-4 still obtains higher accuracy when the length of the chain exceeds the chain length the fine-tuned model saw during its training (>6).
>
> As stated above, however, we believe the strong performance of GPT-4 is due to the simplicity of the axiom. For the d-separation rule, both axiomatically trained small model (67M) and LLama model (8B) show significant improvements over prompting-based GPT-4 (see Tables 3, 4, A6, and A7).
>
> | Model                                   | Length 3 | Length 4 | Length 5 | Length 6 | (OOD-Length) 7 | (OOD-Length) 8 | (OOD-Length) 9 |
> |-----------------------------------------|----------|----------|----------|----------|----------------|----------------|----------------|
> | GPT-4                                   | 0.99     | 0.97     | 0.89     | 0.85     | 0.95           | 0.90           | 0.90           |
> | Gemini Pro                              | 0.75     | 0.73     | 0.72     | 0.76     | 0.71           | 0.68           | 0.74           |
> | Phi-3                                   | 0.88     | 0.86     | 0.82     | 0.79     | 0.76           | 0.73           | 0.79           |
> | Llama-3-8b-Instruct-Finetuned           | 1.00     | 0.98     | 0.93     | 0.86     | 0.85           | 0.76           | 0.71           |
>
> > https://arxiv.org/pdf/2405.15071, discusses the generalization capabilities of large models on synthetic dataset, could be relevant to this work.
>
> Thanks for pointing out this work. We will add a discussion on this, especially in the context of composition-based reasoning.

---

### Official Review · Reviewer_KAqL · 2025-03-16

**Overall Recommendation:** 4

**Summary:**

This paper studies a new method for improving the causal reasoning capabilities of autoregressive transformer text models by training on synthetically generated data containing demonstrations of causal axioms or rules. Specifically, the authors consider the expressions of the form <premise, hypothesis, result> generated using random causal DAGs with structured perturbations. The authors carefully construct the training and test datasets such that the testing measures generalization to unseen causal graphs. By training (or fine-tuning, in case of natural language pre-trained) on transitivity axiom and d-separation rule, and testing on unseen causal graphs as well as causality benchmarks, the authors show that training on linear causal chains and their structural perturbations induces meaningful generalization.

## update after rebuttal

Given the clarifications by the authors, I would like to keep my supportive rating.

**Claims And Evidence:**

The claims made in the submission is supported by clear and convincing evidence. In particular, the authors have made effort in ensuring that genuine generalization capability is measured at test-time, by carefully ensuring that the training data does not contain the test inputs (also for Llama). The evaluation does not only involve the datasets generated by the authors, but also includes two existing causal reasoning benchmarks, which is a good supporting evidence.

**Essential References Not Discussed:**

I am not aware of essential references not discussed in the paper.

**Experimental Designs Or Analyses:**

The experimental design is sound as far as I can verify, especially in the design of the evaluation dataset to be OOD. I have some minor clarifying questions:

- In the beginning of page 5, the authors mention that the training set involves chains with size 3-6 and the evaluation set involves chains of length 5-15. Does this translate to evaluation text inputs (<premise, hypothesis, result>) being longer than the training text inputs?

**Methods And Evaluation Criteria:**

The proposed method and evaluation criteria makes sense for the problem of training a text model to have causal reasoning capabilities as far as I can confirm.

**Other Comments Or Suggestions:**

Minor typos:
- In Lines 60-61, there seems like an un-erased memo.
- In Line 153, "adapted from" is colored blue
- Above Section 4, markov -> Markov
- In Section 7, Refer I -> Refer to I
- In Section 7.1, Refer Table 3 -> Refer to Table 3

**Other Strengths And Weaknesses:**

Strengths

- The paper is well-written and easy to follow.
- The presented evidences on generalization when trained (or fine-tuned) on the proposed synthetic data are strong and significant.

For the weaknesses, I have no major concerns, but would like to hear the authors' response to the following minor concerns and questions.

- The two assumptions employed in this work are the absence of unobserved confounders and the empty conditioning set $Z$. It is relatively less discussed how restrictive these assumptions are.
- While presented in the context of causal reasoning, the axiomatic training and evaluation tasks considered in this paper are essentially same as learning and executing certain graph algorithms (Luca & Fountoulakis, 2024; Sanford et al. 2024; Wang et al. 2023). Some discussion on whether causal reasoning problems are in general a subclass of graph reasoning, or this is a particular characteristic of the problem setup considered in the paper, would be informative.
- The connection to positional encoding and length generalization of transformer in general is interesting, but in the end the paper has resorted to trying out a range of established positional encodings empirically, rather than contributing, e.g., in-depth analysis on why certain positional encodings work better, or a new construction of positional encodings suited for causal reasoning.
- In page 2, the authors mention that the results contribute to the literature on causal learning from passive data; it would be nice to have some discussion on how the paper differs from the prior work in the domain (it is currently missing in Section 2, as far as I can verify).
- The length generalization result was in particular surprising to me, given that some prior work showed that such generalization typically requires some form of process-based training signal (Abbe et al. 2024; Prystawski et al. 2023). Can the authors provide some insights on why the models considered in this work do generalize in length?

Luca & Fountoulakis, Simulation of Graph Algorithms with Looped Transformers (2024)

Sanford et al. Understanding Transformer Reasoning Capabilities via Graph Algorithms (2024)

Wang et al. Can Language Models Solve Graph Problems in Natural Language? (2023)

Abbe et al. How Far Can Transformers Reason? The Globality Barrier and Inductive Scratchpad (2024)

Prystawski et al. Why think step by step? Reasoning emerges from the locality of experience (2023)

**Questions For Authors:**

Please see the weaknesses section.

**Relation To Broader Scientific Literature:**

The contributions in this paper are related to causal reasoning of text models, more generally the out-of-distribution generalization of these models. In the particular context of this paper, the tasks in consideration are essentially reasoning problems on graphs, so there is also a close relation to the capability of text models in learning and executing graph algorithms (please see the Other Strengths And Weaknesses section on this).

**Theoretical Claims:**

The authors do not make theoretical claims in this work.

---

> ### Author Rebuttal · Authors · 2025-04-01
>
> We thank the reviewer for appreciating the contributions of the work. We answer the specific questions below:
>
> **1. Eval inputs being longer than training text inputs?**
>
> Yes the evaluation set consists of chains longer than the ones in the training setup, and this translates to having longer text input than the text inputs in the training set. Table A1 consists of sample instances from our training and evaluation set, showing that the text instances in the evaluation set typically tend to be longer than the ones the model was trained on.
>
> **2. The two assumptions employed...how restrictive these are?**
>
> We assume an empty conditioning set only for the transitivity axiom; for the d-separation rule (Section 3.2), we include conditioning sets of various sizes, as detailed in Section 6.
>
> In both cases, no unobserved confounders were assumed to simplify our setup. If variables' data values are unobserved but the edge structure is known, our framework can readily incorporate unobserved variables symbolically; however, if the structure of unobserved variables is unknown, the problem becomes more complex—a challenge we leave for future work.
>
> **3. Discussion on whether causal reasoning problems are a subclass of graph reasoning**
>
> That's a great point. Our work can be considered as studying a subset of graph algorithms that are relevant for causality. Our first task, transitivity axiom, can be seen as a special case of the graph reachability problem studied in Sanford et al. However, the d-separation task involves a specialized definition for causality that is usually not studied in graph algorithms literature. In other words, our work focuses on the intersection of causality and graph reasoning, specifically on the Pearlian framework of causality which focuses on DAGs. That said, the axiomatic training framework is general and can potentially be extended for other graph reasoning problems, even beyond causality.
>
> **4. The connection to positional encoding .... encodings suited for causal reasoning.**
>
> Our work builds on analyses by Kazemnejad et al. and Shen et al. (references in paper) regarding positional encodings (PEs) and length generalization. While their focus was on tasks like copying and addition, we investigate how different encodings affect length generalization for causal reasoning problems. We corroborate past findings that using NoPE outperforms absolute methods (sinusoidal and learned) because these methods make sequence length explicit during training, leading to poor performance on unseen lengths. In contrast, RoPE improves generalization and addresses these limitations.
> Although designing encodings specifically for causal reasoning would be interesting, our focus is on how axiomatic training aids language models in causal reasoning. Further, we used existing PEs so that our work is applicable to practical models such as Llama. We will clarify these points in the final version.
>
> **5. Prior work on passive learning**
>
> Lampinen et al. examined if agents can develop causal understanding by passively observing interventions on synthetic tasks. Building on their work, our study explores whether causal reasoning axioms can be learned without active interventions by generating passive data from simulated axiom inferences on diverse synthetic graphs. Unlike Lampinen et al., who focus on observational learning for test-time interventions, our approach offers a practical training method—axiomatic training—for language models. This method enhances a transformer's ability to generalize complex causal rules over unseen networks and potentially apply interventional reasoning despite being trained only on passive data. We will add further details in the final draft.
>
> **6. The length generalization result ... compare to past work**
>
> Abbe et al. highlight the “globality barrier,” showing that high-globality tasks like syllogisms require many tokens to capture nontrivial correlations and suggesting scratchpad techniques to break them into manageable subtasks, while Prystawski et al. demonstrate that prompting intermediate steps in locally structured, chain-of-thought reasoning improves performance by decomposing complex inferences into sequential computations. In contrast, our axiomatic training framework constructs a dataset that enables compositional reasoning—such as applying the transitivity axiom repeatedly—by offering **diverse demonstrations**. In line with the claims above, when we only provide sequential causal chains of length 3 as training data, we found that the model did not generalize. We obtain generalizability only when we provide diverse training instances: chains of varying lengths (3–6) and random edge flips, which encourages the model to learn the abstract rule directly rather than relying on intermediate signals. Positional encodings like RoPE further help preserve relative information beyond the training lengths.

---

> > ### Comment · Reviewer_KAqL · 2025-04-03
> >
> > Thank you for providing the clarifications. I would like to keep my supportive rating.

---

### Official Review · Reviewer_DrTc · 2025-03-17

**Overall Recommendation:** 3

**Summary:**

This paper propose an approach where the model learns symbolic axioms through demonstrations rather than directly infer causal relationships from data. And then, they investigate whether this approach allows the model to generalize from learning simple causal structures to more complex causal relationships.

**Claims And Evidence:**

N/A

**Essential References Not Discussed:**

N/A

**Experimental Designs Or Analyses:**

N/A

**Methods And Evaluation Criteria:**

N/A

**Other Comments Or Suggestions:**

N/A

**Other Strengths And Weaknesses:**

Strength
- This paper introduces axiomatic training as a novel approach for training transformers, a field that has not been extensively explored.
- To the best of my knowledge, this is the first work to train transformers specifically to learn causal axioms.
- The proposed method demonstrates generalization to more complex causal relationships beyond the training set.

Weakness

- GPT-4 achieves the best performance, suggesting that causal axioms relation can be learned from unstructured and massive datasets without requiring complex data preprocessing or specialized positional encoding.
- This raises concerns about whether the proposed training approach provides a significant advantage over large-scale unsupervised learning.

**Questions For Authors:**

N/A

**Relation To Broader Scientific Literature:**

I believe this paper makes a meaningful contribution to the scientific leterature.

**Theoretical Claims:**

N/A

---

> ### Author Rebuttal · Authors · 2025-04-01
>
> We thank the reviewer for appreciating the contributions of our work. We answer specific questions below:
>
> **Response to Weaknesses**
> > **1. GPT-4 achieves the best performance, suggesting that causal axioms relation can be learned from unstructured and massive datasets without requiring complex data preprocessing or specialized positional encoding.**
>
> **Response**:
> While GPT-4 performs well on transitivity-based tasks, its accuracy drops to near-random on complex causal rules like d-separation (see Tables A6, A7). Despite its scale, GPT-4 struggles with d-separation, a key rule for inferring conditional independence in causal inference. In contrast, our 67M-parameter model achieves higher accuracy on d-separation, for complex graphs unseen during its training.
>
> Moreover, finetuning Llama-3-8B-Instruct with axiomatic training further improves performance on complex graphs and the CLEAR benchmark (Table 3). On CLEAR’s multiple-choice task, the finetuned model achieves 50% accuracy, outperforming GPT-4’s 36.67% (with a random baseline of around 25%). The CLEAR benchmark has evaluation instances for d-separation that are semantically different from instances in the axiomatic training data. Also, CLEAR benchmark has different types of hypothesis (multiple choice question answer, yes/no type questions), whereas axiomatic instances only contained yes/no hypothesis instances, thus showing potential of the model to generalize for diverse problems beyond simple binary classification.
>
> Lastly, Table 4 shows results on the Corr2Cause benchmark, that evaluates the model's reasoning capability to infer causal relationships from correlational statements, shows the potential of axiomatic training on real world problems which require application of different causality rules (d-separation, transitivity, markov property, markov equivalence class). A Llama-3 model finetuned on axiomatic data obtains 0.37 F1 score and outperforms GPT-4 which struggles to perform well on the task (with 0.29 F1 score).
>
> > **2. This raises concerns about whether the proposed training approach provides a significant advantage over large-scale unsupervised learning.**
>
> **Response**
> We thank the reviewer for this point. While GPT-4 does show potential on simple graph reachability problems requiring application of transitivity axiom, we found that other billion-scale language models (Phi-3 and Gemini Pro) struggle with these problems. Keeping efficiency in mind, we believe our axiomatic framework provides a potential way to improve causal reasoning for smaller models that can balance correctness and efficiency.
>
> Also, as stated above, while GPT-4 model does perform well on transitivity causal axiom, it struggles on applying the d-separation rule of causal inference and performs close to random baseline. This result shows that current billion scale models are unable to reason on fundamental rules of causal inference despite their large scale unsupervised training.

---

### Decision · Program_Chairs · 2025-05-01

**Decision:**

Accept (poster)

**Comment:**

This paper presents a novel training method designed to integrate the transitivity axiom and d-separation rules into autoregressive transformer text models. The authors generate random directed acyclic causal graphs and use these graphs to synthetically create text data in the format of <premise, hypothesis, result>. This approach enables the development of training and testing datasets that allow for the evaluation of the models' generalization capabilities with respect to unseen causal graphs. By training on the transitivity axiom and d-separation rule, and testing on novel causal graphs and benchmarks, the authors demonstrate that training with linear causal chains and their variations results in meaningful generalization.

Overall, the method proposed in this paper is technically sound and reasonable. The simulation results indicate that this approach could enhance the causal reasoning abilities of language models in validating causal hypotheses based on provided causal knowledge. This work has the potential to impact researchers and practitioners in natural language processing and the development of pre-trained text models.

However, the comparison baselines may be biased, as they only include transformer-based text models. There are established proof systems and efficient causal discovery algorithms within the causal inference literature that can infer causal relationships from measured correlations. It would be valuable to compare the proposed text model with these traditional causal inference methods to determine whether high-level causal reasoning capabilities could emerge by incorporating selected causal axioms and d-separation rules.